# Identifying the interplay between protective measures and settings on the SARS-CoV-2 transmission using a Bayesian network

**Pilar Fuster-Parra**[1,2], **Aina Huguet-Torres**[3,4]*, **Enrique Castro-Sánchez**[4,5,6],
**Miquel Bennasar-Veny**[2,3,4,7], **Aina M. Yañez**[2,3,4]

**1** Department of Mathematics and Computer Sciences, University of Balearic Islands, Palma, Spain, **2** Institut
d'Investigació Sanitària Illes Balears (IdISBa), Hospital Universitari Son Espases, Palma, Spain,
**3** Department of Nursing and Physiotherapy, University of Balearic Islands, Palma, Spain, **4** Research Group
on Global Health, University of Balearic Islands, Palma, Spain, **5** College of Business, Arts, and Social
Sciences, Brunel University London, Uxbridge, United Kingdom, **6** Imperial College London, London, United
Kingdom, **7** CIBER de Epidemiología y Salud Pública (CIBERESP), Carlos III Institute of Health (ISCIII),
Madrid, Spain

* aina.huguet@uib.cat

pone.0307041

BRUNEI DARUSSALAM

**Data Availability Statement:** The used datasets
along with the source code are made available

## Abstract

Contact tracing played a crucial role in minimizing the onward dissemination of Severe
Acute Respiratory Syndrome Coronavirus-2 (SARS-CoV-2) in the recent pandemic. Previ-
ous studies had also shown the effectiveness of preventive measures such as mask-wear-
ing, physical distancing, and exposure duration in reducing SARS-CoV-2 transmission.
However, there is still a lack of understanding regarding the impact of various exposure set-
tings on the spread of SARS-CoV-2 within the community, as well as the most effective pre-
ventive measures, considering the preventive measures adherence in different daily
scenarios. We aimed to evaluate the effect of individual protective measures and exposure
settings on the community transmission of SARS-CoV-2. Additionally, we aimed to investi-
gate the interaction between different exposure settings and preventive measures in relation
to such SARS-CoV-2 transmission. Routine SARS-CoV-2 contact tracing information was
supplemented with additional data on individual measures and exposure settings collected
from index patients and their close contacts. We used a case-control study design, where
*close contacts* with a positive test for SARS-CoV-2 were classified as *cases*, and those with
negative results classified as *controls*. We used the data collected from the case-control
study to construct a Bayesian network (BN). BNs enable predictions for new scenarios when
hypothetical information is introduced, making them particularly valuable in epidemiological
studies. Our results showed that ventilation and time of exposure were the main factors for
SARS-CoV-2 transmission. In long time exposure, ventilation was the most effective factor
in reducing SARS-CoV-2, while masks and physical distance had on the other hand a mini-
mal effect in this ventilation spaces. However, face masks and physical distance did reduce
the risk in enclosed and unventilated spaces. Distance did not reduce the risk of infection
when close contacts wore a mask. Home exposure presented a higher risk of SARS-CoV-2
transmission, and any preventive measures posed a similar risk across all exposure settings
analyzed. Bayesian network analysis can assist decision-makers in refining public health

through Zenodo repository at https://doi.org/10.5281/zenodo.10610726.

**Funding:** This study was funded by the Royal College of Nurses from the Balearic Islands (Ref.: 2021-0564). This research was also supported by the Florence Nightingale fellowship program, Royal College of Nurses from the Balearic Islands and the Nursing and Physiotherapy Department, University of the Balearic Islands. The funders had no role in study design, data collection and analysis, decision to publish, or preparation of the manuscript.

**Competing interests:** The authors have declared that no competing interests exist.

campaigns, prioritizing resources for individuals at higher risk, and offering personalized guidance on specific protective measures tailored to different settings or environments.

## Introduction

The novel Severe Acute Respiratory Syndrome Coronavirus-2 (SARS-CoV-2) spread rapidly from Wuhan (China) to the rest of the world, resulting in the coronavirus disease 2019 (COVID-19) pandemic [1, 2]. Essential infection prevention and control measures for respiratory pathogens advocated by the World Health Organisation (WHO) were adopted [3] to contain the dissemination of the infection [4]. Contact tracing, which was adopted by most countries [5, 6], complemented wearing a facemask in public settings, maintaining physical distancing and hand hygiene as main public health approaches to reduce transmission, infection-related hospital admissions, and mortality [7–12]. The measures aimed to mitigate the effect of transmission, such as improving ventilation and minimizing exposure time to infected individuals [13–16], have been identified as essential for controlling community transmission [12, 17]. Even with approaches to reduce the rate of onward transmission, there were moments where such transmission was very high and uncontrollable.

The scientific community has underscored the importance of adhering to preventive measures to mitigate the spread of SARS-CoV-2. However, achieving complete compliance with all recommended measures has proven challenging due to community conditions, environments, and socio-demographic characteristics [18]. Therefore, customizing preventive measures based on the specific feasibility of adoption in each environment could prove advantageous. Further studies and data analysis are necessary to assess the intricate relationship among various preventive factors that influence the transmission of SARS-CoV-2. Concurrently, the dynamics of the pandemic have been significantly shaped by family members, relatives, and close contacts in social and work environments [19]. Despite the substantial contribution of these settings and human networks, the majority of studies have predominantly concentrated on a singular exposure setting, such as hospitals [20–25], households [8, 26], schools [27], or public transportation [28]. Limited research has delved into the transmission of SARS-CoV-2 in other settings where routine activities like working or exercising take place. Additionally, only a small number of studies have considered environmental and exposure-related factors, including ventilation, duration of exposure, and physical distancing [14, 27].

Bayesian networks (BNs) [29, 30] offer a paradigm for interpretable Artificial Intelligence, where high-stakes applications have increased, and therefore, the use of interpretable models is important [31]. In this sense, BNs can be applied to aid health practitioners by providing SARS-CoV-2 characterization estimates as a probability network which updates dynamically as new information is obtained. BNs could also assist in the implementation of most effective prevention measures, regard of which set of measures may be compliance in different every day scenarios [32, 33].

During close contact tracing, carried out at the COVID Coordination Center (CCCOVID) in Mallorca, Spain, additional information was collected to address the lack of evidence regarding the interaction of different exposure settings and preventive measures in relation to SARS-CoV-2 transmission. Therefore, we aimed to evaluate and compare the transmission of SARS-CoV-2 based on individual protective measures and exposure settings. Additionally, we describe the interplay between these prevention measures and the characteristics of exposure settings using a Bayesian network approach with data from a national contact tracing program.

## Materials and methods

### Participants

This study examined a case-control of close contact tracing of SARS-CoV-2-positive patients, detailed in a previously published manuscript [34]. During the data collection period, Mallorca was under the 2nd to 4th level of COVID-19 public health measures (*Control Situation* and *High Risk*). In the food and hospitality sector (i.e., restaurants and bars), up to six people could sit at a table, with social gatherings limited to a maximum of six individuals.

A total of 1, 766 close contacts were included in the sample, of which 420 were positive (cases) and 1, 346 negative (controls). Public and private official testing centers sent daily all positive COVID-19 test results to CCCOVID. Upon receiving this information, nurses contacted the positive patient and did a systematic contact tracing. For this study, positive patients of unknown origin were selected, designating them as index patients. Close contacts were extracted from COVID-19 contact tracing conducted by nurses at the CCCOVID-19 (Balearic Islands). During systematic tracing from February to June 2021, additional information was collected from close contacts of the index patients for this study. A 'close contact' was defined as a person who had been at the same place as an individual with a positive SARS-CoV-2 test, two days before the onset of symptoms or, if the index patient was asymptomatic, from positive diagnosis to isolation of the index patient. In addition, close contacts should have been within 2 meters of the index patient for more than 15 minutes within 24 hours [35]. After systematic close contact tracing, the criteria for selecting the close contact were as follows: asymptomatic and older than 18 years. Exclusion criteria were: 1) to be institutionalized in nursing homes, 2) persons in contact with a healthcare setting and 3) having difficulties with telephone communication or understanding. Inclusion and exclusion criteria for case and controls were the same, except for the SARS-CoV-2 test result, where cases had to be positive (by any diagnostic or screening test used) and controls negative.

All the study procedures followed the Declaration of Helsinki for research on human participants and legal regulations regarding data confidentiality and research involving human participants. The study protocol received approval from the Balearic Committee of Clinical Research Ethics (Ref. no: IB 4444/21). All participants were informed of the study's purpose and procedures before providing their verbal consent to participate.

### Instruments

**Data collection and definition of variables.** Data collection methods are described in detail in the case-control study itself [34]. To summarise, the following data collected regarding the environment or setting, and exposure characteristics associated with SARS-CoV-2 transmission: contact place, ventilation characteristics (open-air, closed space with or without ventilation), mask-wearing, type of mask, duration of contact, shortest distance, case-contact relationship, household members, test result, and handwashing. Social-demographic variables for close contacts were also collected. A description of variables is given in Table 1.

### Learning Bayesian networks

A BN includes [36]: (i) a set of variables or features (nodes) and a set of directed edges between these variables (arcs), (ii) taking into account that each variable or feature (node) has a finite set of mutually exclusive states, and (iii) the variables along with the directed edges (arcs) form a directed acyclic graph (DAG). A BN model estimates the joint probability distribution $P$ over a vector of random variables $\mathbf{X} = (X_1, \ldots, X_n)$. The joint probability distribution, which factorizes as a product of several conditional distributions, represents the dependency/independency

**Table 1. Description of 11 data set variables used to build the BN model.**

| Variable name | Description | Values |
|---|---|---|
| AGEgroup | Age in years | 18-27, 28-42, 43-55, 56-94 |
| EXPOSUREsettings | Exposure place | Education, Home place, Household members, Leisure, Sport place, Transportation, Work place |
| COUGH | Cough present | No, Yes |
| FEVER | Fever present | No, Yes |
| FACEmask | Used Face mask | No, Sometimes, Yes |
| DISTANCE | Physical distance | 0m, 0m-1m, 1m-2m |
| RELATIONSHIP | Relationship | Couples, Friendship, Living family, Other, Other relatives, Work |
| VENTILATION | Type of ventilation | Closed non ventilated, Closed ventilated, Open |
| ExposureTIME | Exposure time | 15min-1h, 1h-4h, 4h-24h, More 24h |
| HandWASHING | Hand hygiene | 1time/day, 2-3 times/day, More 3 times/day |
| SARS-CoV-2 | Infection | No, Yes |

structure by a DAG:

$$P(X_1, \ldots, X_n) = \prod_{i=1}^{n} P(X_i \mid Pa(X_i^{\mathcal{G}}))$$ (1)

Eq (1) (where $Pa(X_i^{\mathcal{G}})$ denotes the parent nodes of $X_i$) which is the main reason for the formulation of a multivariate distribution by BNs; this equation is also called the *chain rule for Bayesian networks*.

The process of learning a BN implies in two tasks: (i) *structural learning*, i.e., consists in identifying the topology (structure defined by a DAG) of the BN, and (ii) *parametric learning*, i.e., estimation of numerical parameters (conditional probabilities assigned to each node of the DAG) given a network topology.

**Structural learning.** Discovering the structure of a BN is a problem where difficulty increases with the number of variables [37].

Three approaches to structure learning could be considered [38]: (i) *search-and-score* structure learning, (ii) *constraint-based* structure learning, and (iii) a combination of both gives a *hybrid* learning framework. *Search-and-score* algorithms assigns a number (score) to each BN structure, and then the structure model with the highest score is chosen. *Constraint-based* search algorithms determine a set of conditional independence exploration on the data. This exploration is used in order to generate an undirected graph. Taking into consideration some additional independence test, the graph is transformed into a BN. *Hybrid algorithms* incorporate characteristics of both *constraint-based* and *score-based* algorithms, they use conditional independence test in order to lessen the search space, and network score in order to obtain the optimal network in the restricted space.

In order to obtain the structure, two possible options can be considered: (i) to select a *single best* model or (ii) to obtain some average model, which is known as *model averaging* [39]. Average models are considered more robust models than the single best ones.

Data were divided into a train data (1,413 close contacts, with 1,010 negative cases and 315 positive cases) to obtain the model and a test data (353 close contacts, with 336 negative cases and 105 positive cases). The *tabu* learning algorithm (a *search-and-score* learning algorithm) from package *bnlearn* [40, 41] of R language [42] was used to learn the structure using a threshold of 0.85 by *model averaging* over 500 networks in order to obtain a more robust model. The score used by the structure learning algorithm was the Akaike Information Criterion (AIC). Prior knowledge of the variables under study was taken into account in order to reduce the number of structures that are consistent with the same set of independencies into the model

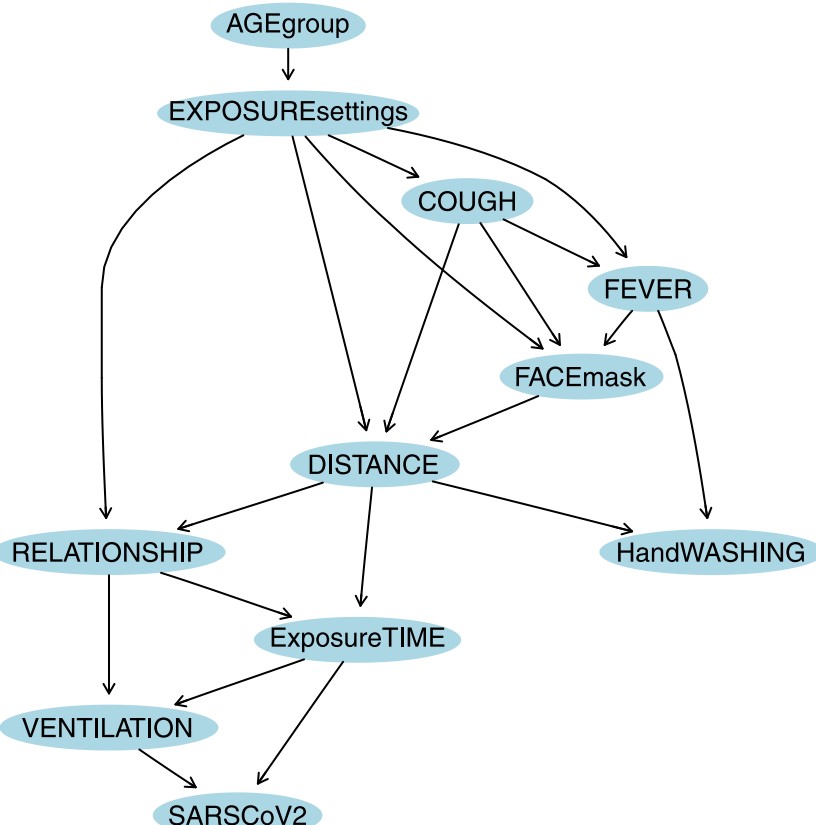

**Fig 1. A possible structure obtained through *tabu* learning algorithm from *bnlearn* package in R language using a threshold = 0.85 by *model averaging* over 500 networks.** Prior knowledge was included in model selection, thus variables were divided into four blocks: 1) *background variables* = {*AGEgroup*, *EXPOSUREsettings*, *COUGH*, *FEVER*}, 2) *conditional variables* = {*DISTANCE*, *HandWASHING*, *FACEmask*}, 3) *intermediate variables* = {*RELATIONSHIP*, *ExposureTIME*, *VENTILATION*}, and, 4) *diagnostic variable* = {*SARS-CoV-2*}.

selection process, and to choose a structure which reflects the causal order and dependencies, causal graphs tend to be sparser [29]. We restrict the model selection process by blacklisting (i, e., disallowing) arrows that point from a later block to an earlier block [43]. An adjacency matrix containing the disallowed edges is constructed, then we convert this into a dataframe.

Variables were divided into four ordered blocks: (i) *background variables* = {*AGEgroup*, *EXPOSUREsettings*, *COUGH*, *FEVER*}, (ii) *conditional variables* = {*DISTANCE*, *HandWASH-ING*, *FACEmask*}, (iii) *intermediate variables* = {*RELATIONSHIP*, *ExposureTIME*, *VENTILA-TION*}, and, (iv) *diagnostic variable* = {*SARS-CoV-2*}.

And therefore, by blacklisting (i,e., disallowing) arrows that point from a later block to an earlier block, arrows are only allowed between variables in the same block and from variables in a block to variables in another block under the first one (from the top to the bottom, i.e, according to the block division: first block is *background variables*, second block is *conditional variables*, third block is *intermediate variables*, and finally fourth block is the *diagnostic variable*).

A DAG was obtained through *tabu* search by model averaging and considering prior knowledge (the causal order given by the four blocks) in model selection (see Fig 1) using *bnlearn* package in R language [42].

**Parametric learning.** In order to obtain the parameters of the BN, a Bayesian parameter estimation through the method *bayes* of *bnlearn* package was used.

**Table 2. Expected values of probabilities for *COUGH* feature conditional on combination of its parent value, in this case conditional on *EXPOSUREsettings* feature.**

| EXPOSUREsettings | COUGH = *No* | COUGH = *Yes* |
|---|---|---|
| Education | 0.4216 | 0.5784 |
| Home place | 0.5151 | 0.4849 |
| Household members | 0.5310 | 0.4690 |
| Leisure | 0.5564 | 0.4436 |
| Sport place | 0.4119 | 0.5881 |
| Transportation | 0.3773 | 0.6227 |
| Work place | 0.4421 | 0.5579 |

A conditional probability distribution was obtained for each node. An example of conditional probability distribution is shown in in Table 2.

## SARS-CoV-2 BN model

Even though the *bnlearn* package [40, 41] in R language [42] could be used to make inference, in order to have a clear graphical representation from the structure and parameters obtained with *bnlearn*, the SARS-CoV-2 BN model was then plotted in Netica [44]. Fig 2 represents the BN once it has been compiled. The joint probability distribution of the SARS-CoV-2 BN in Fig 2 requires the specification of 11 conditional probability tables, one for each variable conditioned to its parents' set.

## Conditional independence

The BN can also be regarded from the perspective that the graph encodes a set of conditional independence assumptions. The *SARS-CoV-2* BN model outputs obtain conditional independencies among the variables. That is, in a BN any node is conditionally independent of its non-descendants given its parents' nodes, i.e. $I(X_i, non - descendants(X)|Pa(X_i))$, as the local Markov property states. In this case, the BN obtained for *SARS-CoV-2* model (see Figs 1 and 2) shows some independencies:

$$I(COUGH, \{AGEgroup\} \mid Pa(COUGH) = \{EXPOSUREsettings\}),$$

$$I(SARS - CoV - 2, \{AGEgroup, EXPOSUREsettings, COUGH, FEVER, \\ HandWASHING, FACEmask, DISTANCE, \\ RELATIONSHIP\} \mid Pa(SARS - CoV - 2) = \\ \{VENTILATION, ExposureTIME\},$$

$$I(FACEmask, \{HandWASHING\} \mid Pa(FACEmask) = \{EXPOSUREsettings, \\ COUGH, FEVER\}),$$

$$I(DISTANCE, \{AGEgroup, FEVER, HandWASHING\} \mid Pa(DISTANCE) = \\ \{EXPOSUREsettings, COUGH, FACEmask\}),$$

$$I(ExposureTIME, \{AGEgroup, EXPOSUREsettings, FEVER, COUGH, \\ HandWASHING\} \mid Pa(ExposureTIME) = \\ \{DISTANCE, RELATIONSHIP, FACEmask\}).$$

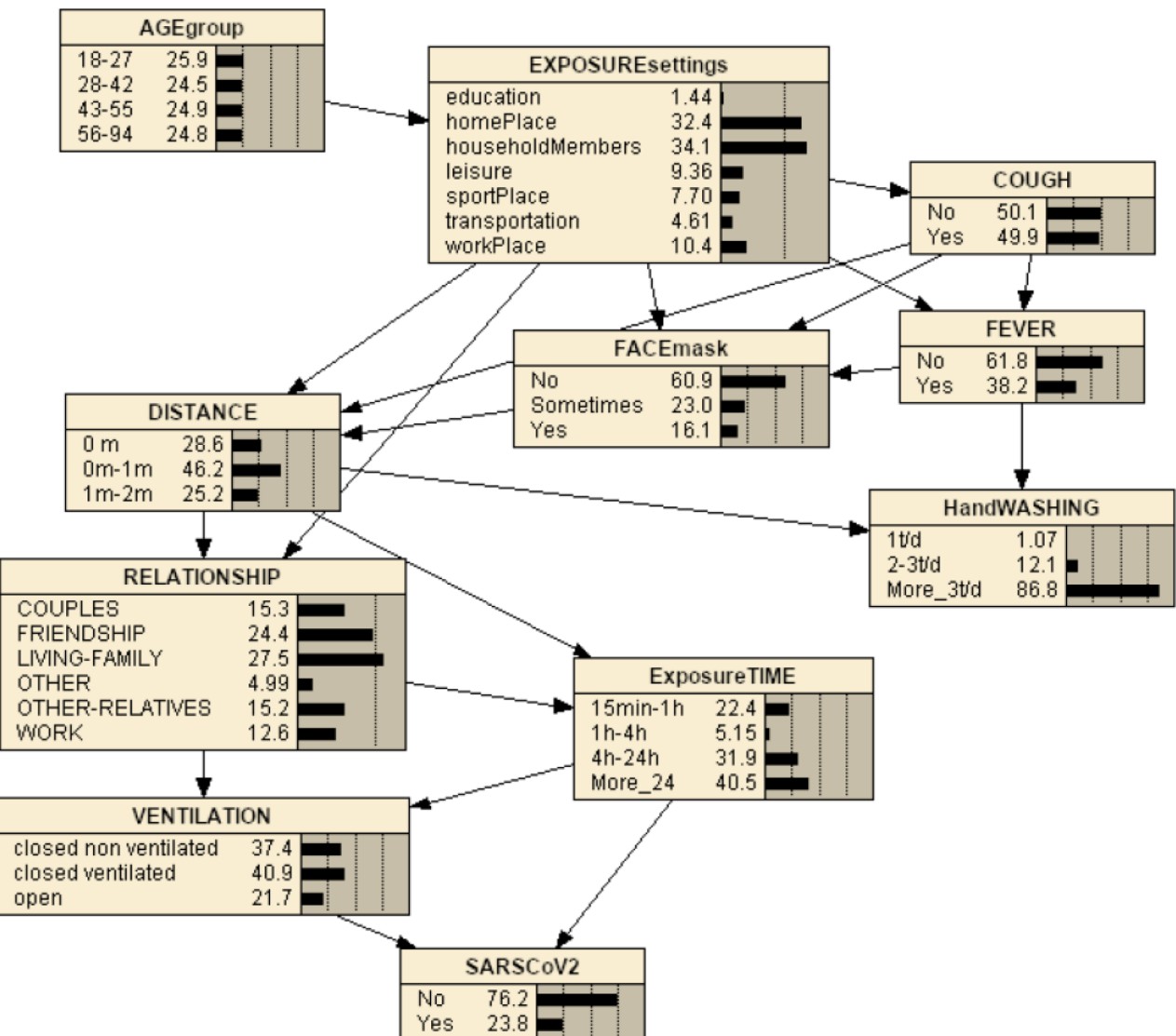

**Fig 2. The compiled SARS-CoV-2 BN model shows a summary of the original data distribution, where probabilities are expressed in percentage.**

## Performance of SARS-CoV-2 BN model

The BN was tested with the test data. In Table 3, the area under the ROC curve (AUC), and the percentage correctly classified for the different features are shown. These last values are obtained with GeNIe software [45].

In addition, three statistics that evaluate the degree of fit compared with a set of new data (the test data): logarithmic loss, quadratic loss and spherical payoff have been considered. Logarithmic loss varies between zero and infinity, where zero indicates the best goodness of fit. The quadratic loss varies between zero and two, where zero corresponds to the best execution, and finally, the spherical payoff is bounded between zero and one, one indicating a perfect fit between the model and the data. In Table 4, we see *SARS-CoV-2* variable has a good degree of fit (logarithmic loss = 0.4817, quadratic loss = 0.3169, spherical payoff = 0.8220).

**Table 3. AUCs and percentage correctly classified for the different features.**

| Variable name | State | AUC | Accuracy |
|---|---|---|---|
| AGEgroup | 18-27 | 0.6208 | 34.72 |
| AGEgroup | 28-42 | 0.5829 | 34.72 |
| AGEgroup | 43-55 | 0.5629 | 34.72 |
| AGEgroup | 56-94 | 0.6132 | 34.72 |
| EXPOSUREsettings | Education | 0.9887 | 49.81 |
| EXPOSUREsettings | Home place | 0.7041 | 49.81 |
| EXPOSUREsettings | Household members | 0.7265 | 49.81 |
| EXPOSUREsettings | Leisure | 0.7701 | 49.81 |
| EXPOSUREsettings | Sport place | 0.8733 | 49.81 |
| EXPOSUREsettings | Transportation | 0.8030 | 49.81 |
| EXPOSUREsettings | Workplace | 0.8316 | 49.81 |
| COUGH | No | 0.6781 | 62.11 |
| COUGH | Yes | 0.6781 | 62.11 |
| FEVER | No | 0.6612 | 65.36 |
| FEVER | Yes | 0.6612 | 65.36 |
| FACEmask | No | 0.7907 | 67.70 |
| FACEmask | Sometimes | 0.7530 | 67.70 |
| FACEmask | Yes | 0.8177 | 67.70 |
| DISTANCE | 0m | 0.9135 | 67.47 |
| DISTANCE | 0m-1m | 0.7735 | 67.47 |
| DISTANCE | 1m-2m | 0.8437 | 67.47 |
| RELATIONSHIP | Couples | 0.9486 | 55.17 |
| RELATIONSHIP | Friendship | 0.8156 | 55.17 |
| RELATIONSHIP | Living family | 0.7256 | 55.17 |
| RELATIONSHIP | Other | 0.8999 | 55.17 |
| RELATIONSHIP | Other relatives | 0.7794 | 55.17 |
| RELATIONSHIP | Work | 0.8644 | 55.17 |
| VENTILATION | Closed non ventilated | 0.7585 | 58.71 |
| VENTILATION | Closed ventilated | 0.6656 | 58.71 |
| VENTILATION | Open | 0.8568 | 58.71 |
| ExposureTIME | 15min-1h | 0.7773 | 57.43 |
| ExposureTIME | 1h-4h | 0.8193 | 57.43 |
| ExposureTIME | 4h-24h | 0.7187 | 57.43 |
| ExposureTIME | More 24h | 0.8688 | 57.43 |
| HandWASHING | 1t/day | 0.7314 | 86.79 |
| HandWASHING | 2-3t/day | 0.5957 | 86.79 |
| HandWASHING | More 3t/day | 0.5864 | 86.79 |
| SARS-CoV-2 | No | 0.7466 | 76.23 |
| SARS-CoV-2 | Yes | 0.7466 | 76.23 |

**Performance comparison.** We report other classification performances (see Table 5) in order to provide reference benchmarks about how our BN classifies, only for informational purposes, as the main objective of the study was on modelling a potential BN model for studying SARS-CoV-2 transmission. We include the widely used: Support vector machine (SVM), Multilayer Perceptron (MLP), and the Random Forest algorithms integrated in WEKA [46]. SVM fins a hyperplane that best separates data into different classes while maximizing the

**Table 4. Scoring results obtained with Netica.**

| Variable name | Logarithmic loss | Quadratic loss | Spherical payoff |
|---|---|---|---|
| AGE | 1.3690 | 0.7379 | 0.5104 |
| COUGH | 0.7311 | 0.5001 | 0.7095 |
| DISTANCE | 0.9438 | 0.5549 | 0.6683 |
| EXPOSUREsettings | 1.3910 | 0.6544 | 0.5879 |
| ExposureTIME | 1.1400 | 0.5768 | 0.6381 |
| FACEmask | 0.8342 | 0.4665 | 0.7191 |
| FEVER | 0.6562 | 0.4440 | 0.7454 |
| HandWASHING | 0.3752 | 0.2037 | 0.8929 |
| RELATIONSHIP | 1.3380 | 0.6229 | 0.6052 |
| VENTILATION | 0.8996 | 0.5344 | 0.6821 |
| SARS-CoV-2 | 0.4817 | 0.3169 | 0.8220 |

margin between them, and can handle non-linear relationships through the use of kernel functions. Multilayer Perceptron (MLP) is a type of neural network with multiple layers, including input, hidden, and output layers. And, Random Forest builds multiple decision trees during training and merges them together to get a more accurate and stable prediction, it is less prone to overfitting compared to individual decision trees.

Only the diagnostic feature (*SARS-CoV-2*) was considered as a comparative example. Performance of each classification model is evaluated using three statistical measures: accuracy, sensitivity, specificity and the ROC area. Accuracy represents the ratio of the number of correct predictions to the total number of predictions. Sensitivity represents the true positive rate, while specificity represents the true negative rate. The ROC (Receiver Operating Characteristics) or relative operating curves is a graphical plot of the sensitivity. The ROC area or AUC (Area under the Curve) is defined as the probability of correctly classifying a pair of cases (positive and negative), and it is used as a predictive indicator of goodness.

From Tables 3–5 we conclude that the SARS-CoV-2 BN provides a computational efficient prediction system for the study of the relationships of different aspects related to SARS-CoV-2 infection.

## Results

### MAP queries

BNs allow the calculation of new probabilities once new information is inputted [47]. An important task consists of finding a high-probability joint assignment to a subset of features

**Table 5. Performance for *SARS-CoV-2* feature comparing our BN with the corresponding algorithms integrated in Weka.**

| Algorithms | Accuracy | Sensitivity | Specificity | ROC Area |
|---|---|---|---|---|
| Bayesian network | 76.2264 | 0.8512 | 0.4857 | 0.7466 |
| Lib SVM | 79.5918 | 0.7960 | 0.3990 | 0.5980 |
| Multilayer Perceptron | 67.8005 | 0.6780 | 0.4210 | 0.6370 |
| Random Forest | 74.1497 | 0.7410 | 0.4150 | 0.6660 |

Accuracy $= \frac{TP+TN}{P+N}$; Sensitivity $= \frac{TP}{TP+FN}$; Specificity $= \frac{TN}{TN+FP}$. Where: True Positive (TP), False Negative (FN), False Positive (FP), True Negative (TN), Positive (P), Negative (N).

[29]. A variant of this type of task is the *most probable explanation* (also known as MAP query). The goal is to find the most likely assignment to the features in $W = complementary(E)$ given the evidence $E = e$:

$$MAP(W \mid e) = argmax_w P(w, e) \qquad (2)$$

The most likely assignment to a single feature in a *probability query*, that is, to compute $P(X|e)$ is taking into account. The most likely joint assignment to $W$ in a MAP query is found. In order to analyze the BN, the concept of Markov blanket of a node (which includes its parents, its children, and the children's other parents (spouses)) together with causal (where we predict effects from causes (we proceed from top to bottom)) and intercausal (which constitutes a very common pattern in human reasoning, when different causes of the same effect can interact) reasoning patterns were considered.

**Minimizing *SARS-CoV-2* in *yes* state.** The initial probability of being infected by *SARS-CoV-2* is of 23.80% expressed in percentage, and we are interested in knowing the variables' states that minimize that value with different type of ventilation: open, closed ventilated, and closed non ventilated. For that, in order to determine the influence of the different values states of *VENTILATION* feature on the evolution of *SARS-CoV-2* feature in *yes* state (with an initial probability of 23.80% expressed in percentage), different features are instantiated (they take a value), i.e., *evidence* is introduced, *VENTILATION* in their three possible states (step 1), followed of *FACEmask* feature in *yes* state (step 2), then *DISTANCE* in *1m-2m* (step 3), and finally *ExposureTIME* feature in *15m-1h* state (step 4). We observe that, when *ExposureTIME* feature is instantiated (in step 4), then the *FACEmask* and *DISTANCE* features do not have any influence on *SARS-CoV-2* feature, the trail of influence is broken (the Markov blanket of *SARS-CoV-2* feature is composed by *ExposureTIME* and *VENTILATION* features (its parents, its children, and the other parents of its children or espouses)). As expected, under these instantiations, the estimated probability of *SARS-CoV-2* feature in *yes* state decreases (achieving an estimated probability of 2.92% when *VENTILATION* is open, 4.99% when *VENTILATION* is closed ventilated, and 18.10% when *VENTILATION* is closed non ventilated, from the initial value of 23.80% once the BN is compiled). A summary of the propagation of influences can be seen in Fig 3. Also, the results showed that in open spaces the mask and the distance do not lower the risk of contagion, but the time of exposure does. However, these measures do lower the risk in closed ventilated or unventilated spaces.

**Face mask influence.** The use of facemasks is evaluated in *SARS-CoV-2* propagation when *DISTANCE* is instantiated to *1m-2m* (the maximum distance considered in this study), *ExposureTIME* is instantiated to *15m-1h* (the minimum exposure time considered in this study), and finally *HandWASHING* feature is instantiated to state *more 3 times*. As seen in Fig 4 the use of face masks was not relevant when *DISTANCE* was maximum and *ExposureTIME* minimum. Hand hygiene does not show any influence on *SARS-CoV-2* feature, at any level for *FACEmask* feature. Keeping the maximum prevention distance and minimum exposure time shows a clear decrease on *SARS-CoV-2* feature in *yes* state (achieving an estimated conditional probability of 7.49% when *FACEmask* is instantiated to *yes* state, 7.55% when *FACEmask* is instantiated to *sometimes* state, and 7.75% when *FACEmask* is instantiated to *no* state). Distance did not reduce the risk of infection when close contacts wore a mask. However, regardless of mask usage and maximum distance, minimizing exposure time resulted in a similar decrease in risk at any level.

**Influence of exposure settings under negative measures.** The influence of the different exposure settings on *SARS-CoV-2* variable at the value *yes* is evaluated under negative measures (where the estimated conditional probabilities once *EXPOSUREsettings* variable is

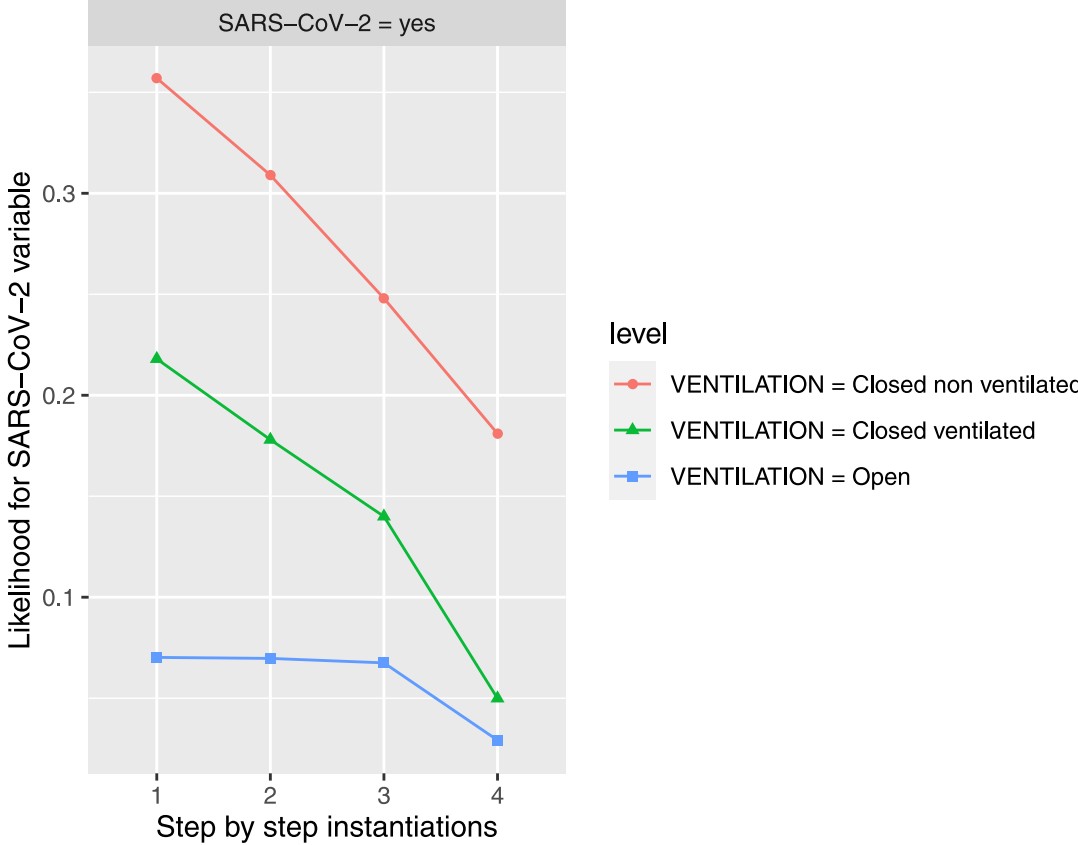

**Fig 3. The probability of infection by SARS-CoV-2 is updated in the different steps when new evidence is introduced for the three possibilities of ventilation.** The different steps: step 1 = *VENTILATION*, step 2 = *FACEmask* in state *yes*, step 3 = *DISTANCE* in state *1m-2m*, and step 4 = *ExposureTIME* in state *15m-1h*, to evaluate *SARS-CoV-2* feature in *yes* state. The steps are represented in the horizontal axis, while the estimated probability for *SARS-CoV-2* variable at the value *yes* is shown in the vertical axis.

instantiated gives 17.70% for *education*, 23.40% for *home place*, 28.20% for *household members*, 19.90% for *leisure*, 17.00% for *sport place*, 20.40% for *transportation*, and 21.60% for *work place*, expressed in percentage).

According to that, *FACEmask* variable is instantiated to *no* state, *VENTILATION* variable is instantiated to *closed non ventilated* state, and finally *DISTANCE* variable is instantiated to *0 m* state, showing a similar conditional probability estimation for *SARS-CoV-2* feature (see Fig 5) for the different exposure settings (37.70% for education, 43.30% for home place, 43.30% for household members, 42.70% for leisure, 37.60% for sport place, 41.50% for transportation, and 42.40% for work place, expressed in percentage), which increases under the instantiations of negative measures, although the differences of estimated conditional probabilities are shorten. Home exposure had a higher risk of SARS-CoV-2 transmission, but not taking any prevention measures posed a similar risk across all exposure settings analyzed.

**Influence of exposure time under positive measures.** The *ExposureTIME* variable is considered under the different states to evaluate their influence on *SARS-CoV-2* feature in *yes* state, showing little difference under positive measures (*FACEmask* variable in state *yes*, *DISTANCE* variable in state *1m-2m*, and *VENTILATION* in state *open*), as it is showed in Fig 6. Observe that once *VENTILATION* is instantiated to *open* in step 4, then *FACEmask* and

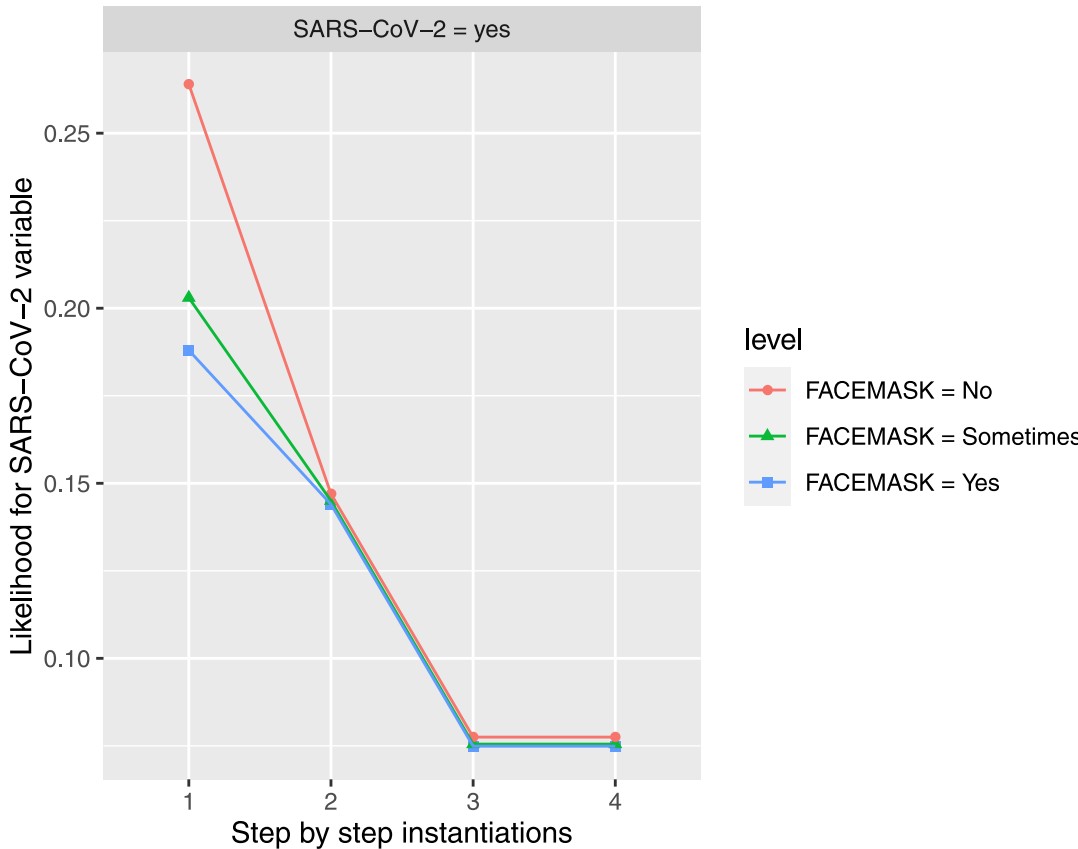

**Fig 4. The probability of infection by SARS-CoV-2 is updated in the different steps when new evidence is introduced for the three possibilities of using face mask.** The different steps: step 1 = *FACEmask*, step 2 = *DISTANCE* in state *1m-2m*, step 3 = *ExposureTIME* in state *15m-1h*, and step 4 = *HandWASHING* in state *more 3 times*, to evaluate *SARS-CoV-2* feature in *yes* state. The different steps are represented in the horizontal axis, while the estimated probability for *SARS-CoV-2's* variable at the value *yes* is shown in the vertical axis.

*DISTANCE* do not have any influence on *SARS-CoV-2* feature, as any trail would be broken (the Markov blanket of *SARS-CoV-2* feature (*VENTILATION* and *ExposureTIME* features) is instantiated), however, we consider this order in the instantiations to see their influence (The estimated conditional probability for *SARS-CoV-2* feature in *yes* state, at the last step is 2.92% for *ExposureTIME* at *15m-1h* state, 9.68% for *ExposureTIME* at *1h-4h* state, 9.44% for *ExposureTIME* at *4h-24h* state, 8.48% for *ExposureTIME* at *more 24h* state). VENTILATION variable was an effective factor in reducing SARS-CoV-2 infection risk in exposure time level, while masks and maintaining distance have minimal effect.

## Discussion

Our study has shown the suitability of BNs in epidemiological research. BN allowed us to quantify the effect of protective measures and the exposure setting on SARS-CoV-2 transmission.

Characterising the interplay between different factors that contribute to SARS-CoV-2 transmission through model development and analysis can have both theoretical and practical benefits, for example in explanation, prediction, monitoring, and prevention of any emerging and non-emerging diseases. BN models are especially well-suited for investigating the risk factors

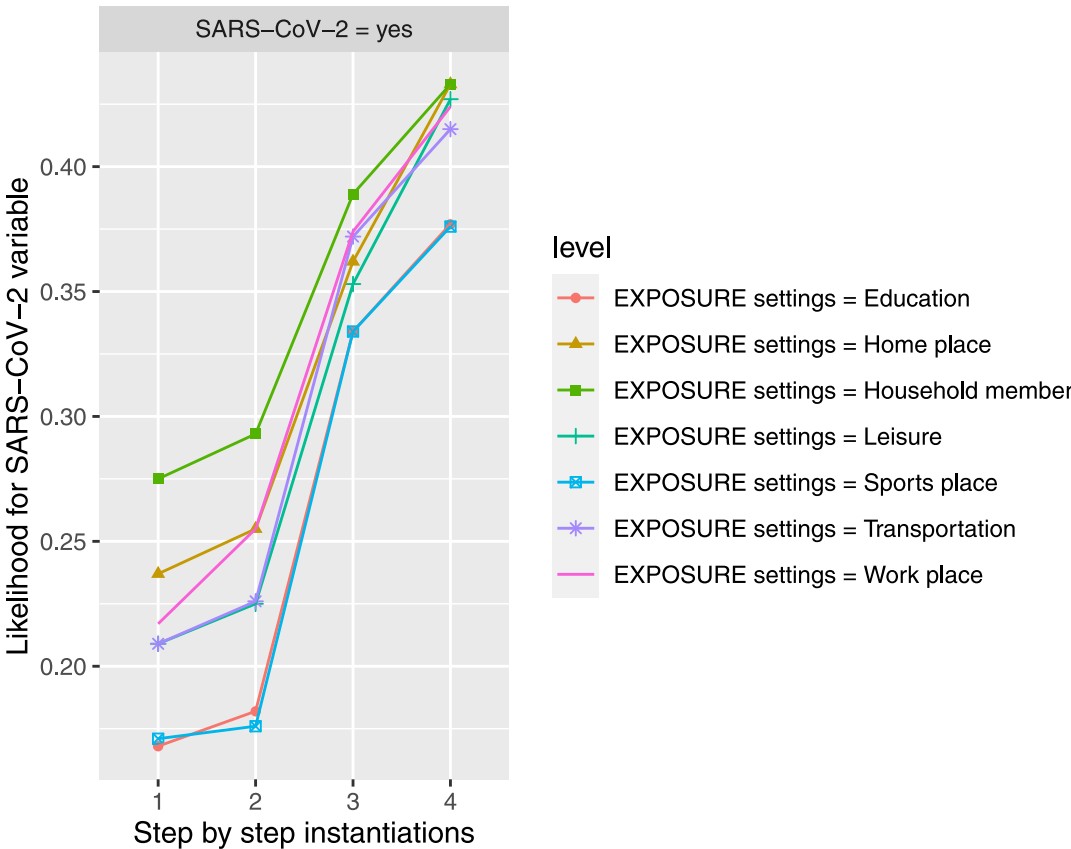

**Fig 5. The probability of infection by SARS-CoV-2 is updated in the different steps when new evidence is introduced for the seven possibilities of exposure place.** The different steps: step 1 = *EXPOSUREsettings*, step 2 = *FACEmask* in state *no*, step 3 = *VENTILATION* in state *closed non ventilated*, and step 4 = *DISTANCE* in state *0 m*, to evaluate *SARS-CoV-2* feature in *yes* state. The different steps are represented in the horizontal axis, while the estimated probability for *SARS-CoV-2's* variable at the value *yes* is shown in the vertical axis.

for SARS-CoV-2 transmission in the community since they can analyse intricate relationships between multiple variables and account for the probabilistic nature of causal dependencies [4, 48]. Additionally, BN models can create hypothetical scenarios based on new observations, which would facilitate knowledge mobilization and optimal/effective decision-making. Moreover, the utilization of the SARS-CoV-2 transmission model has the potential to offer a thorough description of all the variables included in the analysis, providing a complete understanding of the relationships between them.

This case-control study evaluates the risk factors for infection among the close contacts of an index patient diagnosed with SARS-CoV-2 of unknown origin, using a BN model and the Markov blanket concept. Our results showed that maintaining contact in an open or ventilated space, exposure time, mask use and distance from the index patient are determinants for SARS-CoV-2 transmission. Furthermore, limited exposure time with the index patient is an important protective factor even in the presence of other protective factors such as being outdoors and wearing a mask. The results of our analysis are in accordance with previous evidence reporting that exposure time is the main important risk factor for SARS-CoV-2 transmission [14, 23, 49].

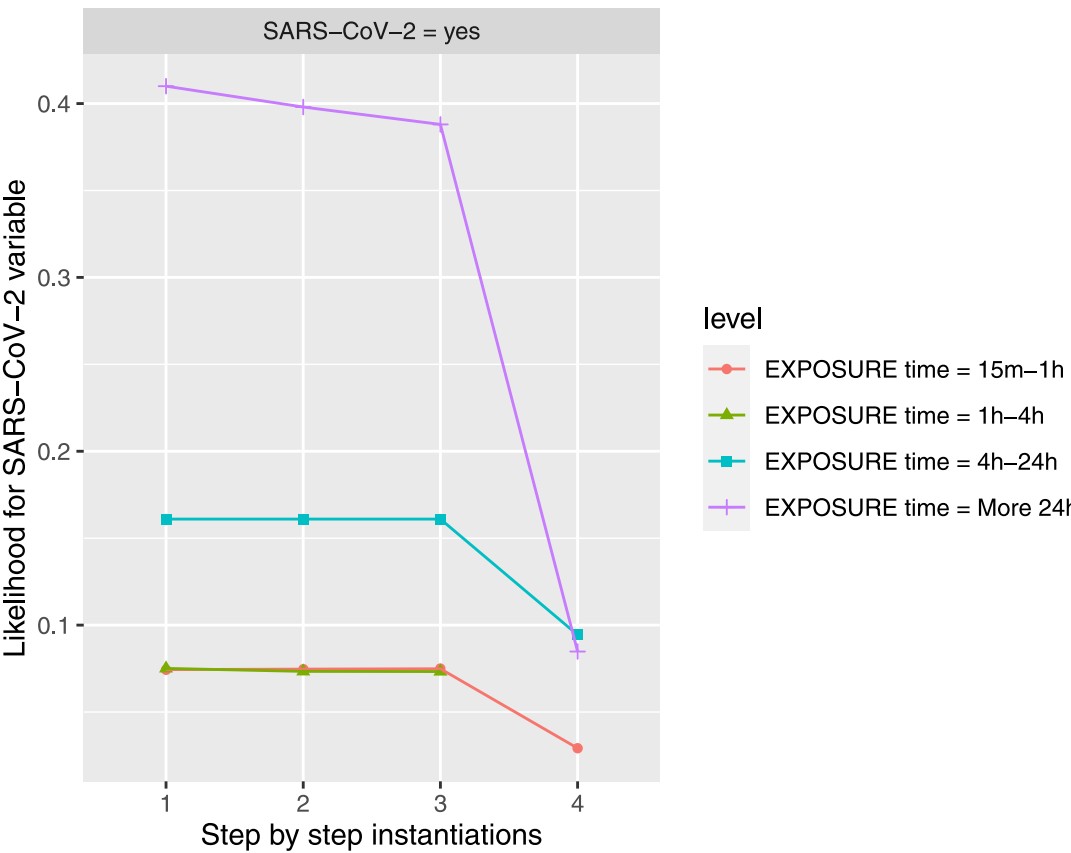

**Fig 6. The probability of infection by SARS-CoV-2 is updated in the different steps when new evidence is introduced for the four possibilities of exposure time.** The different steps: step 1 = *ExposureTIME*, step 2 = *FACEmask* in state *yes*, step 3 = *DISTANCE* in state *1m-2m*, and step 4 = *VENTILATION* in state *open*, to evaluate *SARS-CoV-2* feature in *yes* state. The different steps are represented in the horizontal axis, while the estimated probability for *SARS-CoV-2's* variable at the value *yes* is shown in the vertical axis.

The primary distinction from other research on SARS-CoV-2 and transmission are that this study utilized intercausal reasoning and the notion of a Markov blanket to enhance the contact tracing feature. The reason for choosing the BN model is that it generates probability estimates rather than predictions. Unlike regression, the process of learning the structure of a BN is a type of unsupervised learning where the model does not differentiate between the dependent variable and the independent ones. This is advantageous because it avoids the limitations of a traditional regression model.

This study also indicates that the use of a facemask would reduce the risk of transmission when a close contact occurs in an unventilated, enclosed space but not in the open air. Previous studies have reported similar results about the higher risk of transmission in poorly ventilated spaces [13, 50], but have however not compared the use of masks and ventilation. In addition, our results showed that for close contacts who do not wear a mask, maintaining a safe distance from others would have a comparable effect in terms of risk of infection as wearing a mask. Although, a lot of studies reported lower SARS-CoV-2 transmission using masks and maintaining distance [8, 11, 51] few studies analysed the combination of both measures. Our study showed that social distancing can help to reduce the risk of transmission even in situations where wearing a mask were difficult. A previous study reported results, showing a

synergistic effect between use of facemasks and social distancing [52]. While wearing a mask remains an important factor, the distance effect could complement wearing mask measures to reduce the risk of transmission.

Moreover, our research implies that while living index patient poses the greatest risk of infection, not implementing preventive measures in other exposure settings can also lead to a similar risk regardless of how long the exposure lasts. Various studies have revealed similar findings, with households having a higher percentage of secondary attack rates compared to other exposure settings [53–56]. For instance, one study demonstrated how wearing masks and maintaining social distance within households can make a significant difference [11]. These results highlight the significance of implementing protective and preventive measures during at-risk contact, rather than relying on the link between the index patient and their close contacts.

## Limitations

Our study has limitations in terms of its design and methods. Firstly, we assumed that all positive contacts had been infected by the index patient, whereas some contacts may have contracted the virus from other unknown sources in the community. For that reason, there could be misclassification of exposure settings and prevention measures, potentially leading to reduced statistical power. Nevertheless, our results have demonstrated significant differences.

Another limitation of the study is that the variability in the types of masks used by the participants was not thoroughly analyzed, which could have influenced the results.

Compared with other types of analysis, such as logistic regression that also use the outcome as a binary, BN analysis evaluates the risk of different settings, adding the presence of different variables (ventilation, face mask, etc.) instead of adjusting for the effect of other factors as logistic regression [34] that evaluate the adjusted effect of each condition. BN modeling is a more practical approach for risk management since provides probability estimates for different scenarios that could easily interpret.

Our study has several strengths. Firstly, we included the majority of index patients diagnosed between February and June 2021, ensuring a comprehensive sample size. Secondly, we analyzed all COVID tests conducted in Mallorca during the study period, providing results that can be generalized to the wider community. Also to minimize the bias of case-control studies [57], the questionnaire was administered before the respondents were notified of their test results, ensuring that their responses were not influenced to minimize the complacency. Additionally, recall bias was minimized by contacting individuals at the time of their identification as close contacts, minimizing the time elapsed between the exposure and the interview.

## Conclusion

Ventilation and exposure time are the main risk factors for SARS-CoV-2 transmission while wearing a mask in unventilated spaces is an important protective factor in this setting. The BN analysis is an advanced model for dynamic description and prediction to identify risk exposure settings and the most effective personal protective measures at each setting. In addition, the BNs tool could help public institutions to focus on individuals most at-risk and disseminate community messages with more targeted and compelling recommendations in future pandemic scenarios. Furthermore, by understanding the specific protective measures that are applicable in different settings or environments, more personalized guidance could be provided to individuals.

## Acknowledgments

We thank all participants involved in providing information. We thank all professionals from COVID-19 Coordinating Centre for their support, especially Rocío Sánchez Rodríguez, Ana Belén Expósito Torres, Esther Granados Ramos, María José Sastre Perea.

## Author Contributions

**Conceptualization:** Aina Huguet-Torres, Enrique Castro-Sánchez, Miquel Bennasar-Veny, Aina M. Yañez.

**Data curation:** Aina Huguet-Torres.

**Formal analysis:** Pilar Fuster-Parra, Enrique Castro-Sánchez, Miquel Bennasar-Veny, Aina M. Yañez.

**Funding acquisition:** Aina Huguet-Torres.

**Methodology:** Pilar Fuster-Parra, Miquel Bennasar-Veny, Aina M. Yañez.

**Supervision:** Aina M. Yañez.

**Writing – original draft:** Pilar Fuster-Parra, Aina Huguet-Torres.

**Writing – review & editing:** Enrique Castro-Sánchez, Miquel Bennasar-Veny, Aina M. Yañez.

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
