## [Decision Letter · Decision Letter 0]

5 Jun 2024

PONE-D-24-17062Identifying the interplay between protective measures and settings on the SARS-CoV-2 transmission using a Bayesian networkPLOS ONE

Dear Dr. Huguet-Torres,

Thank you for submitting your manuscript to PLOS ONE. After careful consideration, we feel that it has merit but does not fully meet PLOS ONE’s publication criteria as it currently stands. Therefore, we invite you to submit a revised version of the manuscript that addresses the points raised during the review process.

We look forward to receiving your revised manuscript.

Kind regards,

Liling Chaw

Academic Editor

PLOS ONE

Journal Requirements:

   "This study was funded by the Royal College of Nurses from the Balearic Islands (Ref.: 2021-0564). This research was also supported by the Florence Nightingale fellowship program, Royal College of Nurses from the Balearic Islands and the Nursing and Physiotherapy Department, University of the Balearic Islands."

Additional Editor Comments:

Please consider the comments from the reviewers.

Reviewers' comments:

Reviewer's Responses to Questions

**Comments to the Author**

1. Is the manuscript technically sound, and do the data support the conclusions?

Reviewer #1: Yes

Reviewer #2: Yes

2. Has the statistical analysis been performed appropriately and rigorously? 

Reviewer #1: Yes

Reviewer #2: Yes

3. Have the authors made all data underlying the findings in their manuscript fully available?

Reviewer #1: Yes

Reviewer #2: Yes

4. Is the manuscript presented in an intelligible fashion and written in standard English?

Reviewer #1: Yes

Reviewer #2: Yes

5. Review Comments to the Author

Reviewer #1: Very nice work, I find the topic and the bayesian network approach really interesting and very well presented. I've just highlighted some minor comments.

Abstract

Very well written

Introduction

L.10 - 12 Sentence to be revisited, difficult to follow

L.28 - 30 References missing

L.43 Additionally, describe -> something is missing here (e.g., we describe / our work describes)

Materials and Methods

L. 46 Maybe number M&M as 2 and Introduction as 1 (do the same for the next sections - Results, Discussion)

L.130-133 - Please check the names of the variables to be consistent in the text

L.193-195 Please provide more info what Accuracy, Sensitivity Specificity, ROC Area describe?

Results + Discussion

Very nicely reported

Reviewer #2: A very good piece of work. I liked the way you showed the structure of the variables in terms of the BN construction. I think it is a worthwhile addition to the literature, the way the underlying data is used supports the objectives of the paper well. Cross comparisons with other classifiers are helpful in highlighting the strong applicability of BN to this type of problem. I thought you were also honest and realistic about the limits of contact tracing, as a process and the likelihood of infection from the index case, vs contamination from another source.

The presence of the handwashing as a non-causal variable was an excellent way to validate both the resulting model and the process. I felt you could have emphasised this more strongly.

One aspect that could have been explored was the variability in masks; i.e. the mask type. I was disappointed that although this information was mentioned there was no further exploration. Other studies have shown the different effectiveness of mask types, as well as the expected leakage for infected persons. The limited effectiveness of certain mask types (surgical vs FFP2/N95) against aerosol transmission in poorly ventilated spaces has been documented elsewhere, perhaps your dataset wasn’t large enough or lacked the necessary details to clearly illustrate this signal.

Another point which would have benefited from clarification was the status of education and workplace activities during the period of the study (i.e. restrictions, working from home, school or university closures). I assumed in my review that there were no restrictions/lockdown measures in place for work or education in the study region during the period when the data was collected.

6. PLOS authors have the option to publish the peer review history of their article (what does this mean?). If published, this will include your full peer review and any attached files.

Reviewer #1: No

Reviewer #2: No

---

## [Author Response · Author response to Decision Letter 0]

14 Jun 2024

Dear Editor-in-Chief, 

Thank you for the opportunity to submit a revised draft of our manuscript titled "Identifying the interplay between protective measures and settings on the SARS-CoV-2 transmission using a Bayesian network” to PLOSONE. 

We appreciate the time and effort that you and the reviewers have dedicated to providing valuable feedback to our manuscript. We are grateful to the reviewers for their insightful comments, which have strengthened the manuscript. We have highlighted the changes within the manuscript, and our responses to the reviewers below in blue colour. We uploaded the revised version of the manuscript with tracked changes and a clean version. 

#Journal Requirements:

R: Thank you for the feedback. The manuscript has been reviewed and revised to ensure it meets PLOS ONE's style requirements, including file naming conventions.

R: Thank you for your guidance regarding the handling of funding information. We have promptly addressed this matter by removing all funding-related text from the manuscript, as per your instruction.

 "This study was funded by the Royal College of Nurses from the Balearic Islands (Ref.: 2021-0564). This research was also supported by the Florence Nightingale fellowship program, Royal College of Nurses from the Balearic Islands and the Nursing and Physiotherapy Department, University of the Balearic Islands."

R: Thank you for your suggestion. The funders had no role in the study design, data collection and analysis, decision to publish, or preparation of the manuscript. We will include this statement in the cover letter of the online submission form.

R: Thank you for your comment. As you suggested, the data has been made available.

R: All references have been reviewed, and no retracted references have been identified. 

Reviewers' comments:

Reviewer's Responses to Questions

Comments to the Author

1. Is the manuscript technically sound, and do the data support the conclusions?

Reviewer #1: Yes

Reviewer #2: Yes

R: Thank you. We appreciate your decision.

2. Has the statistical analysis been performed appropriately and rigorously?

Reviewer #1: Yes

Reviewer #2: Yes

R: Thank you. We appreciate your decision.

3. Have the authors made all data underlying the findings in their manuscript fully available?

Reviewer #1: Yes

Reviewer #2: Yes

R: Thank you. We appreciate your decision.

4. Is the manuscript presented in an intelligible fashion and written in standard English?

Reviewer #1: Yes

Reviewer #2: Yes

R: Thank you. We appreciate your decision.

5. Review Comments to the Author

Reviewer #1: Very nice work, I find the topic and the bayesian network approach really interesting and very well presented. I've just highlighted some minor comments.

Abstract

Very well written

R: Thank you for your comment.

Introduction

L.10 - 12 Sentence to be revisited, difficult to follow

R: Thank you for your suggestion. The sentence has been modified for improved clarity. The new sentence reads: 

‘The measures aimed at mitigating the effects of transmission, such as improving ventilation and minimizing exposure time to infected individuals (13-16), have been identified as essential for controlling community transmission (12,17).’

L.28 - 30 References missing

R: Thank you for bringing this to my attention. We apologize for the oversight. We have now added the missing references to the manuscript. 

L.43 Additionally, describe -> something is missing here (e.g., we describe / our work describes)

R: Thank you for detecting this error. We modified the sentence. The new sentence reads:

‘Additionally, we describe the interplay between these prevention measures and the characteristics of exposure settings using a Bayesian network approach with data from a national contact tracing program.’

Materials and Methods

L. 46 Maybe number M&M as 2 and Introduction as 1 (do the same for the next sections - Results, Discussion)

R: Thank you for your suggestion regarding section numbering. However, upon reviewing the journal guidelines, we have decided to remove section numbering as it does not align with the journal's policies. 

L.130-133 - Please check the names of the variables to be consistent in the text

R: Thank you for the observation. We have checked the names of variables in the text.

L.193-195 Please provide more info what Accuracy, Sensitivity Specificity, ROC Area describe?

R: Thank you for the suggestion. We have added some information related to these concepts in the text.

Results + Discussion

Very nicely reported

R: Thank you for your comment.

 

Reviewer #2: A very good piece of work. I liked the way you showed the structure of the variables in terms of the BN construction. I think it is a worthwhile addition to the literature, the way the underlying data is used supports the objectives of the paper well. Cross comparisons with other classifiers are helpful in highlighting the strong applicability of BN to this type of problem. I thought you were also honest and realistic about the limits of contact tracing, as a process and the likelihood of infection from the index case, vs contamination from another source. The presence of the handwashing as a non-causal variable was an excellent way to validate both the resulting model and the process. I felt you could have emphasised this more strongly.

R: Thank you for your comments. As you rightly pointed out, the inclusion of handwashing, or other variables, with non-causal outcomes has underscored the significance of all other preventive and/or protective measures. We appreciate your observation and the value you have added to the study.

One aspect that could have been explored was the variability in masks; i.e. the mask type. I was disappointed that although this information was mentioned there was no further exploration. Other studies have shown the different effectiveness of mask types, as well as the expected leakage for infected persons. The limited effectiveness of certain mask types (surgical vs FFP2/N95) against aerosol transmission in poorly ventilated spaces has been documented elsewhere, perhaps your dataset wasn’t large enough or lacked the necessary details to clearly illustrate this signal.

R: We appreciate your observation regarding mask types, and we regret that this aspect was not thoroughly explored in the present article. In a previous publication, based on the same sample, the type of mask did not have significant results once adjusted for other variables. Additionally, most participants reported using surgical masks, as per official guidance at the time, which influenced our decision not to include this factor in the Bayesian network model. In our context, and in the social situations in non-clinical environments, FFP2/NP95 face masks were seldom used. We acknowledge though that including information about the type of marks could have contributed to the body of evidence about mask effectiveness. We will therefore include this aspect in the limitations of this study.

The new sentence reads: 

‘Another limitation of the study is that the variability in the types of masks used by the participants was not thoroughly analyzed, which could have influenced the results.’

 

Another point which would have benefited from clarification was the status of education and workplace activities during the period of the study (i.e. restrictions, working from home, school or university closures). I assumed in my review that there were no restrictions/lockdown measures in place for work or education in the study region during the period when the data was collected.

R: We appreciate your observation regarding the status of education and workplace activities during the study period. We regret the oversight in not providing a detailed clarification in the methodology section of the article. While there were no restrictions on education or workplace activities during the data collection period, there were limitations in other areas, such as bars and social settings. We acknowledge that this aspect could have been elucidated more comprehensively to ensure a thorough understanding of the study context. It's important to note that this information is covered in detail in another article. Furthermore, to address this concern and enhance the clarity and coherence of our work, we will include a concise explanation of the restrictions applicable at that time in the relevant section of the methodology. This addition will facilitate a better understanding of the environmental conditions in which our study was conducted.

The new sentence reads: 

‘During the data collection period, Mallorca was under the 2nd to 4th level of COVID-19 public health measures ("Control Situation" and "High Risk"). In the food and hospitality sector (ie, restaurants and bars), up to six people could sit at a table, with social gatherings limited to a maximum of 6 individuals.’

---

## [Decision Letter · Decision Letter 1]

28 Jun 2024

Identifying the interplay between protective measures and settings on the SARS-CoV-2 transmission using a Bayesian network

PONE-D-24-17062R1

Dear Dr. Huguet-Torres,

We’re pleased to inform you that your manuscript has been judged scientifically suitable for publication and will be formally accepted for publication once it meets all outstanding technical requirements.

Kind regards,

Liling Chaw

Academic Editor

PLOS ONE

Additional Editor Comments (optional):

Well done!

Reviewers' comments:

Reviewer's Responses to Questions

**Comments to the Author**

1. If the authors have adequately addressed your comments raised in a previous round of review and you feel that this manuscript is now acceptable for publication, you may indicate that here to bypass the “Comments to the Author” section, enter your conflict of interest statement in the “Confidential to Editor” section, and submit your "Accept" recommendation.

Reviewer #1: All comments have been addressed

Reviewer #2: All comments have been addressed

2. Is the manuscript technically sound, and do the data support the conclusions?

Reviewer #1: Yes

Reviewer #2: Yes

3. Has the statistical analysis been performed appropriately and rigorously? 

Reviewer #1: Yes

Reviewer #2: Yes

4. Have the authors made all data underlying the findings in their manuscript fully available?

Reviewer #1: Yes

Reviewer #2: Yes

5. Is the manuscript presented in an intelligible fashion and written in standard English?

Reviewer #1: Yes

Reviewer #2: Yes

6. Review Comments to the Author

Reviewer #1: All my comments have been addressed. Again I find the topic and the bayesian network approach really interesting very well presented.

Reviewer #2: All comments addressed in the manuscript or adequately explained in the Author's response letter. The raw data has been made available (checked on Zenodo).

7. PLOS authors have the option to publish the peer review history of their article (what does this mean?). If published, this will include your full peer review and any attached files.

Reviewer #1: **Yes: **Eleftherios Meletis

Reviewer #2: No

---

## [Editor Report · Acceptance letter]

3 Jul 2024

PONE-D-24-17062R1 

PLOS ONE

Dear Dr. Huguet-Torres, 

I'm pleased to inform you that your manuscript has been deemed suitable for publication in PLOS ONE. Congratulations! Your manuscript is now being handed over to our production team.

Kind regards, 

on behalf of

Dr. Liling Chaw 

Academic Editor

PLOS ONE